# A Polar Robust Kalman Filter Algorithm for DVL-Aided SINSs Based on the Ellipsoidal Earth Model

**DOI:** 10.3390/s22207879

**Published:** 2022-10-17

**Authors:** Ming Tian, Zhonghong Liang, Zhikun Liao, Ruihang Yu, Honggang Guo, Lin Wang

**Affiliations:** 1College of Advanced Interdisciplinary Studies, National University of Defense Technology, Changsha 410073, China; 2College of Intelligence Science and Technology, National University of Defense Technology, Changsha 410073, China; 3Nanhu Laser Laboratory, National University of Defense Technology, Changsha 410073, China

**Keywords:** AUV, polar navigation, DVL-aided SINS, Mahalanobis distance, robust Kalman filter

## Abstract

Autonomous underwater vehicles (AUVs) play an increasingly essential role in the field of polar ocean exploration, and the Doppler velocity log (DVL)-aided strapdown inertial navigation system (SINS) is widely used for it. Due to the rapid convergence of the meridians, traditional inertial navigation mechanisms fail in the polar region. To tackle this problem, a transverse inertial navigation mechanism based on the earth ellipsoidal model is designed in this paper. Influenced by the harsh environment of the polar regions, unknown and time-varying outlier noise appears in the output of DVL, which makes the performance of the standard Kalman filter degrade. To address this issue, a robust Kalman filter algorithm based on Mahalanobis distance is used to adaptively estimate measurement noise covariance; thus, the Kalman filter gain can be modified to weight the measurement. A trial ship experiment and semi-physical simulation experiment were carried out to verify the effectiveness of the proposed algorithm. The results demonstrate that the proposed algorithm can effectively resist the influence of DVL outliers and improve positioning accuracy.

## 1. Introduction

Recently, the importance of the polar region, especially the Arctic Ocean, in traffic, economic, military, and scientific research has received increasing attention. Many researchers are devoted to polar science. In order to better explore the polar region, many challenging problems need to be solved. Polar navigation is the primary issue to be addressed. No matter what kind of carrier (e.g., the autonomous underwater vehicle (AUV)), precise navigation information is essential for the performance of tasks in the polar region. Considering the flexibility and concealment of AUVs, they can break through the season and terrain constraints of the polar environment for detection activities, improve data acquisition efficiency, and realize intelligent polar environment detection [1]. Thus, AUVs play an important role in polar ocean exploration, and AUV related navigation research is receiving increasing focus [2,3,4]. The reliability and accuracy of AUV navigation systems is critical for AUVs accomplishing tasks autonomously.

However, compared to the navigation method used in non-polar regions, the polar navigation of AUVs faces some difficulties. First, due to the peculiarities of polar geography, the traditional strapdown inertial navigation system (SINS) mechanism in the local-level geographic frame loses efficacy in the polar region [5]. The reason is that the meridian converges rapidly as the latitude increases, and the north reference definition thus becomes meaningless. Moreover, because the command angular velocity includes the tangent of latitude, calculation overflow and error amplification are inevitable in the polar region. A lot of work has been devoted to solving this problem. The main idea is to redefine the navigation frame used in the polar region in order to solve the calculation overflow problem and the north reference failure issue. Among the redefined navigation frames, the grid frame and transverse frame are generally used for polar navigation [6,7]. The grid frame adopts the intersection of the grid plane, which is parallel to the Greenwich plane and the tangent plane as grid north, thus the meridian rapid convergence in the polar region is avoided. The SINS mechanism in the grid frame has been widely adopted by many aircrafts because of its adaptability with aerial flight charts. The transverse frame redefines the transverse earth frame and the transverse local-level geographic frame through special rotation from the traditional earth frame and local-level geographic frame. The polar region in the traditional local-level geographic frame becomes adjacent to the equator in the transverse local-level geographic frame. Its north reference definition and position representation method also vary with changes in the navigation frame. The transverse frame is widely used in marine navigation. Yao et al. [8] proposed a transverse frame navigation algorithm and verified the scope of application of the transverse frame, demonstrating that the transverse frame at the mid-latitudes has accuracy as consistent as the traditional local-level geographic frame. However, the traditional transverse frame definition is based on the spherical earth model which has an error in principle.

Secondly, because of the complexity of the natural polar environment, the use of common navigation systems (e.g., the global navigation satellite system (GNSS) or the geomagnetic navigation system) is limited in the polar region, especially for the underwater polar environment. Though the SINS, which is the major navigation equipment for AUVs, has the advantages of autonomy and concealment, its navigation errors (caused by inertial sensor errors) accumulate over time due to its dead reckoning principle. Thus, the SINS cannot work alone for extended periods of time. To overcome this problem, an aided navigation system must be introduced to assist the SINS in achieving high-precision navigation. Considering navigation accuracy and the concealment requirement, the Doppler velocity log (DVL) is ideal equipment for assisting an AUV’s SINS [9,10]. The DVL transmits ultrasonic waves from the hull to the seabed or water layer and receives the reflected signal, with which it can calculate the speed of the AUV in real-time according to the Doppler effect. It can provide real-time precise external speed observation for the SINS, and the navigation computer can fuse the navigation information from the SINS and the DVL through a filter algorithm, such as the Kalman filter, the extended Kalman filter, the unscented Kalman filter, and the particle filter.

As a whole, the DVL-aided SINS has become the mainstream solution for current underwater navigation [11], and it is also an important way of realizing polar navigation with AUVs. Zhang et al. [12] designed a DVL-aided SINS navigation algorithm in the transverse frame, which can effectively suppress the increase of azimuth misalignment angle in the polar region. However, due to the influence of ice layers, the underwater acoustic environment in the polar region is more complicated than the non-polar region. Moreover, the other unknown factors, e.g., the unknown polar ocean current, also influence the working status of DVLs. The operation of the DVL may be unstable, and the abnormal values may pollute the DVL measurement values. Additionally, the mentioned phenomenon is more common in the polar region than in the non-polar region, which can significantly affect the accuracy of the DVL-aided SINS. Aiming at solving this problem, an adaptive Kalman filter or robust Kalman filter is usually adopted [13,14,15,16,17]. The main idea of the adaptive Kalman filter is to adjust the measurement noise covariance matrix or the system noise covariance matrix, which can further modify the Kalman filter gain and improve filter performance. Many studies have been published. Sun et al. [18] proposed an adaptive Kalman filter based on the interactive dual model. They derived a method to dynamically estimate the noise covariance matrix, which enhances the accuracy of the navigation system in the Arctic region. However, the algorithm does not fundamentally solve the problem regarding failure of the local-level geographic frame in the polar region. Though the algorithm proposed by Zhang et al. [12] shows effectiveness in polar DVL-aided SINSs, the algorithm does not consider the influence of the environment on DVL measurement noise, so its performance in terms of robustness is more or less poor. Chang [19] proposes a robust Kalman filter algorithm, with the Mahalanobis distance between the measurement vector and its one-step prediction as a criterion, which can resist the influence of outliers on the filter results and adaptively estimate the measurement noise covariance value. The algorithm has good performance in terms of improving navigation accuracy in the presence of observation outliers.

In this paper, a new transverse integrated navigation method based on the earth ellipsoidal model is proposed. Compared to the traditional method, the proposed method reduces the error model based on the earth ellipsoid model and has higher accuracy in relation to long-endurance navigation. In addition, a DVL-aided SINS using a robust Kalman filter based on the Mahalanobis distance is proposed in order to cope with the complex environment in polar regions. The ship experiments and semi-physical simulation experiments verify that the algorithm can improve the positioning accuracy of the navigation system at mid-latitudes and in the polar region and resist the impact of DVL outliers on positioning accuracy. The structure of this paper is as follows: Section 2 introduces the definition of the transverse frame and designs a corresponding SINS mechanism based on the earth ellipsoidal model. Section 3 analyzes the system dynamic model of the DVL-aided SINS and designs a respective measurement model. In Section 4, a robust Kalman filter with Mahalanobis distance as a criterion is proposed to resist the influence of outliers on the filter results and adaptively estimate the measurement noise covariance value. In Section 5, the ship and semi-physical experiments used to evaluate the proposed algorithm are described. Finally, conclusions are drawn in Section 6.

## 2. Transverse Frame Definition and SINS Navigation Mechanism

The transverse frame can be used as navigation frame in the polar region. The meridian convergence at high latitudes has no effect on it due to the fact that the polar region in the traditional local-level geographic frame becomes the region adjacent to the equator in the redefined transverse frame. This section introduces the definition of the transverse frame, including the transverse earth frame and transverse local-level geographic frame, as well as the transformation relationship between them. Further, based on assumption of the ellipsoidal earth model, the SINS mechanism in the transverse local-level geographic frame is designed.

### 2.1. Transverse Frame Definition

There are several different transverse frame definitions, and the differences between them are small. The transverse frame defined in this paper is the same as the one defined in Yao et al. [8]. As shown in Figure 1, the transverse earth frame is denoted by the *e*’ frame. The *e*’ frame can be defined through two sequence rotations of the earth-centered earth-fixed frame, i.e., *e* frame. Namely, the *e* frame is rotated −90° around its *x*_e_ axis, then rotated −90° around the intermediate *z*_e_ axis. The direction cosine matrix (DCM) between them can be written as follows:(1)$$C_{e}^{e^{\prime }}=\left[\begin{array}{ccc} \cos(-90^{\circ }) & \sin(-90^{\circ }) & 0\\ -\sin(-90^{\circ }) & \cos(-90^{\circ }) & 0\\ 0 & 0 & 1 \end{array}\right]\left[\begin{array}{ccc} 1 & 0 & 0\\ 0 & \cos(-90^{\circ }) & \sin(-90^{\circ })\\ 0 & -\sin(-90^{\circ }) & \cos(-90^{\circ }) \end{array}\right]=\left[\begin{array}{ccc} 0 & 0 & 1\\ 1 & 0 & 0\\ 0 & 1 & 0 \end{array}\right]$$

The Greenwich meridian plane becomes the transverse equator in the transverse earth frame. The intersection point of the 90° E meridian with the equator is the transverse north pole N’, and the intersection point of the 90° W meridian with the equator is the transverse south pole S’. The transverse parallel is parallel to the transverse equator. The semi-ellipse passing through the transverse north pole *N*’, the transverse south pole *S*’, and the north pole N is the transverse 0° meridian. The transverse local-level geographic frame is defined as the *t*-frame. Its definition is as follows: the origin is at point *P_0_*, which is the projection point of the AUV position point *P* on the local-level plane; the *x_t_* axis is along the tangent of the transverse parallel and toward to the transverse east; the *z_t_* axis is along the upward direction, which is vertical to the local-level plane, and coincides with the *z_g_* axis of the local-level geographic frame; the *y_t_* axis, *x_t_* axis, and *z_t_* axis constitute a right-handed orthogonal frame. The *x_t_* axis, *y_t_* axis, and *z_t_* axis are respectively denoted by *E^t^*, *N^t^*, and *U^t^*. The projection of point *P* on the transverse equator is denoted by point *M*. The intersection point of the normal *PP_0_* and the transverse equator is denoted by point *Q*. The transverse longitude *λ^t^* is defined as the included angle between the transverse equator and the *PQM* plane. The transverse latitude *L^t^* is defined as the included angle between the line *PQ* and the transverse equator, namely, the included angle between the line *PQ* and the line *MQ*.

Moreover, in this paper, the *i* frame, *b* frame, *d* frame, and the *g* frame respectively represent the inertial frame, the body frame of the AUV, the body frame of the DVL, and the local-level geographic frame, with tri-axes denoted by *E^g^*, *N^g^*, *U^g^* (*x_g_* axis, *y_g_* axis, and *z_g_* axis). The *b* frame and *d* frame both adopt the ‘right-front-up’ definition.

The DCM from the *e* frame to the *g* frame is written as
(2)$$C_{e}^{g}=\left[\begin{array}{ccc} -\sin\lambda & \cos\lambda & 0\\ -\sin L\cos\lambda & -\sin L\sin\lambda & \cos L\\ \cos L\cos\lambda & \cos L\sin\lambda & \sin L \end{array}\right]$$
where *L* is the traditional latitude and *λ* the traditional longitude.

Similarly, the DCM from the *e*’ frame to the *t* frame is written as
(3)$$C_{e^{\prime }}^{t}=\left[\begin{array}{ccc} -\sin\lambda^{t} & \cos\lambda^{t} & 0\\ -\sin L^{t}\cos\lambda^{t} & -\sin L^{t}\sin\lambda^{t} & \cos L^{t}\\ \cos L^{t}\cos\lambda^{t} & \cos L^{t}\sin\lambda^{t} & \sin L^{t} \end{array}\right]$$

According to the chain rule, the DCM from the *g* frame to the *t* frame can be written as
(4)$$C_{g}^{t}=C_{e^{\prime }}^{t}C_{e}^{e^{\prime }}C_{g}^{e}$$

By geometric trigonometry, the relationships between the transverse latitude and transverse longitude and the traditional latitude and traditional longitude are as follows:(5)$$\begin{array}{l} L^{t}=\arctan\left(\frac{\cos L\sin\lambda }{\sqrt{1-\cos^{2}L\sin^{2}\lambda }}\right)=\arccos\left(C_{23}\right)\\ \lambda^{t}=\arctan\left(\cot L\cos\lambda \right)=\arccos\left(C_{12}\right) \end{array}$$
where $C_{23}$ represents the element in the second row and third column of the matrix $C_{e^{\prime }}^{t}$.$C_{12}$ represents the element in the first row and second column of the matrix $C_{e^{\prime }}^{t}$. The inverse transformation relationship between them can also be obtained by a similar method. Note that the height definition *h^t^*, no matter whether in the transverse local-level geographic frame or the local-level geographic frame, is the same and is defined as the vertical distance from the horizontal plane. The height’s positive direction is upward.

According to the above defined transverse local-level geographic frame, there is an angle, σ, between the transverse north reference and the traditional north reference. The DCM from the *g* frame to the *t* frame can also be written as
(6)$$C_{g}^{t}=\left[\begin{array}{ccc} \cos\sigma & -\sin\sigma & 0\\ \sin\sigma & \cos\sigma & 0\\ 0 & 0 & 1 \end{array}\right]$$

According to Equations (4) and (6), substituting Equations (1)–(3) and (5) into Equation (4) yields
(7)$$\begin{array}{l} \sin\sigma =\frac{\cos\lambda }{\sqrt{1-\cos^{2}L\sin^{2}\lambda }}=\frac{\sin L^{t}}{\sqrt{1-\cos^{2}L^{t}\cos^{2}\lambda^{t}}}\\ \cos\sigma =\frac{-\sin L\sin\lambda }{\sqrt{1-\cos^{2}L\sin^{2}\lambda }}=\frac{-\sin L^{t}\cos\lambda^{t}}{\sqrt{1-\cos^{2}L^{t}\cos^{2}\lambda^{t}}} \end{array}$$

### 2.2. Transverse SINS Mechanism Based on the Earth Ellipsoidal Model

Generally, the transverse SINS mechanism adopts the spherical earth model regarding the flexibility of mathematical calculation. However, the earth is closer to an ellipsoid, and there is an error in principle related to the spherical assumption. Aiming at solving this problem, the earth ellipsoidal model is adopted in this paper. In fact, under assumption of the ellipsoidal earth model, the transverse north reference can be obtained by rotating the grid north reference −90° around the upward direction, which makes it easier to transform between them.

The attitude update equation in the transverse local-level geographic frame is written as
(8)$${\dot{C}}_{b}^{t}=C_{b}^{t}\left[\omega_{ib}^{b}\times \right]-\left[\omega_{it}^{t}\times \right]C_{b}^{t}$$
where $C_{b}^{t}$ is the DCM from the *b* frame to the *t* frame, $\omega_{ib}^{b}$ is the angular velocity of the *b* frame relative to the *i* frame as measured by the gyro assembly, [∙×] is the symmetric matrix of vector ∙, and $\omega_{it}^{t}$ is the angular velocity of the *t* frame relative to the *i* frame, which can be expressed as
(9)$$\omega_{it}^{t}=\omega_{ie^{\prime }}^{t}+\omega_{e^{\prime }t}^{t}$$
with
(10)$$\omega_{ie^{\prime }}^{t}=C_{e^{\prime }}^{t}\omega_{ie^{\prime }}^{e}=C_{e^{\prime }}^{t}\left[\begin{array}{c} \omega_{ie}\\ 0\\ 0 \end{array}\right],\;\omega_{e^{\prime }t}^{t}=\left[\begin{array}{cc} \frac{1}{\tau } & -\frac{1}{R_{y}}\\ \frac{1}{R_{x}} & -\frac{1}{\tau }\\ \frac{\tan L^{t}}{R_{x}} & -\frac{\tan L^{t}}{\tau } \end{array}\right]\left[\begin{array}{c} v_{E}^{t}\\ v_{N}^{t} \end{array}\right]$$
where $\omega_{ie^{\prime }}^{t}$ is the angular velocity of the *e*’ frame relative to the *i* frame, $\omega_{e^{\prime }t}^{t}$ is the angular velocity of the *t* frame relative to the *e*’ frame, $\omega_{ie}$ is the earth rotation rate, *R_x_* is the radius of the curvature along the *x* axis, *R_y_* is the radius of the curvature along the *y* axis, *τ* is the twist rate of the ellipsoidal, $v_{E}^{t}$ is the transverse east velocity, and $v_{N}^{t}$ is the transverse north velocity. The right side of Equation (10) can be written as
(11)$$\begin{array}{l} \frac{1}{R_{x}}=\frac{\sin^{2}\sigma }{R_{N}+h^{t}}+\frac{\cos^{2}\sigma }{R_{E}+h^{t}},\;\frac{1}{R_{y}}=\frac{\cos^{2}\sigma }{R_{N}+h^{t}}+\frac{\sin^{2}\sigma }{R_{E}+h^{t}}\\ \frac{1}{\tau }=\left(\frac{1}{R_{N}+h^{t}}-\frac{1}{R_{E}+h^{t}}\right)\sin\sigma \cos\sigma \\ R_{E}=\frac{R_{e}}{\sqrt{1-e^{2}\cos^{2}L^{t}\cos^{2}\lambda^{t}}}\\ R_{N}=\frac{R_{e}{\left(1-e\right)}^{2}}{{\left(1-e^{2}\cos^{2}L^{t}\cos^{2}\lambda^{t}\right)}^{3/2}} \end{array}$$
where $R_{e}$ is the earth semi-major axis radius and e is the eccentricity of the earth.

The velocity update equation in the transverse local-level geographic frame is written as
(12)$${\dot{v}}^{t}=C_{b}^{t}f^{b}-\left(2\omega_{ie^{\prime }}^{t}+\omega_{e^{\prime }t}^{t}\right)\times v^{t}+g^{t}$$
where $v^{t}={\left[\begin{array}{ccc} v_{E}^{t} & v_{N}^{t} & v_{U}^{t} \end{array}\right]}^{T}$ is the AUV’s velocity, $f^{b}$ is the specific force measured by the accelerometer assembly, and $g^{t}$ is the gravity vector.

The position evolution in the transverse local-level geographic frame is expressed as the following position DCM differential equation and height differential equation:(13)$$\begin{array}{l} {\dot{C}}_{e^{\prime }}^{t}=-\left[\omega_{e^{\prime }t}^{t}\times \right]C_{e^{\prime }}^{t}\\ {\dot{h}}^{t}=v_{U}^{t} \end{array}$$

When the AUV sails into the polar region, the navigation frame is switched from the *g* frame to the *t* frame. Therefore, the navigation parameters should also be projected in the corresponding navigation frame, and the transformation relationship is as follows:(14)$$\begin{array}{l} C_{b}^{t}=C_{g}^{t}C_{b}^{g}\\ v^{t}=C_{g}^{t}v^{g}\\ C_{e^{\prime }}^{t}=C_{g}^{t}C_{e}^{g}C_{e^{\prime }}^{e}\\ h^{t}=h \end{array}$$

Similarly, when the AUV sails out of the polar region, the inverse transformation relationship can be obtained.

## 3. Kalman Filter Model Design for a DVL-Aided SINS in the Transverse Frame

With the assist of the DVL, the SINS error can be restrained, and the polar navigation accuracy can thus be maintained. Generally, a Kalman filter is used as the information fusion method to fuse the information from the DVL and the SINS. This section introduces the system dynamic model and measurement model of the DVL-aided SINS, and corresponding models are designed in the transverse local-level geographic frame.

### 3.1. System Dynamic Model Design

The system dynamic model of the DVL-aided SINS includes two parts: the SINS error equation and the DVL error equation. The SINS error equation includes the attitude error equation, the velocity error equation, and the position error equation. These equations have similar styles to the corresponding equations projected in the traditional local-level geographic frame. These equations can be obtained by perturbing Equations (8), (12), and (13).

The SINS attitude error equation projected in the *t* frame is as follows:(15)$${\dot{\phi }}^{t}=-\omega_{it}^{t}\times \phi^{t}+\delta \omega_{it}^{t}-C_{b}^{t}\delta \omega_{ib}^{b}$$
with
(16)$$\delta \omega_{it}^{t}=\delta \omega_{ie^{\prime }}^{t}+\delta \omega_{e^{\prime }t}^{t},\;\delta \omega_{ib}^{b}=\varepsilon^{b}+w_{g}^{b}$$
(17)$$\delta \omega_{ie^{\prime }}^{t}=\delta C_{e\prime }^{t}\omega_{ie^{\prime }}^{e}=-\left[\delta \theta^{t}\times \right]C_{e^{\prime }}^{t}\omega_{ie^{\prime }}^{e}$$
(18)$$\delta \omega_{e^{\prime }t}^{t}=\left[\begin{array}{cc} 0 & -\frac{1}{R_{eh^{t}}^{}}\\ \frac{1}{R_{eh^{t}}^{}} & 0\\ \frac{\tan L^{t}}{R_{eh^{t}}^{}} & 0 \end{array}\right]\left[\begin{array}{c} \delta v_{E}^{t}\\ \delta v_{N}^{t} \end{array}\right]+\left[\begin{array}{ccc} 0 & 0 & \frac{v_{N}^{t}}{R_{eh^{t}}^{2}}\\ 0 & 0 & -\frac{v_{E}^{t}}{R_{eh^{t}}^{2}}\\ \frac{v_{E}^{t}}{R_{eh^{t}}^{}\cos^{2}L^{t}} & 0 & -\frac{v_{E}^{t}\tan L^{t}}{R_{eh^{t}}^{2}} \end{array}\right]\left[\begin{array}{c} \delta \theta_{E}^{t}\\ \delta \theta_{N}^{t}\\ \delta h^{t} \end{array}\right]$$
where $\delta \theta^{t}$ is the position error angle related to the position DCM $C_{e^{\prime }}^{t}$ (its components are $\delta \theta_{E}^{t}$, $\delta \theta_{N}^{t}$, and $\delta \theta_{U}^{t}$, respectively), $\delta \omega_{ib}^{b}$ is the gyro assembly error including gyro bias $\varepsilon^{b}$ and gyro noise $w_{g}^{b}$, and $R_{eh^{t}}=R_{e}+h^{t}$ is just a simplified representation. Note that the Equation (18) ignores the earth ellipse and the influence of distortion. $\delta \theta_{E}^{t}$ and $\delta \theta_{N}^{t}$ are the equivalent representational forms of position error.

The SINS velocity error equation projected in the *t* frame is as follows:(19)$$\delta {\dot{v}}^{t}=f^{t}\times \phi^{t}-\left(2\omega_{ie^{\prime }}^{t}+\omega_{e^{\prime }t}^{t}\right)\times \delta v^{t}+v^{t}\times \left(2\delta \omega_{ie^{\prime }}^{t}+\delta \omega_{e^{\prime }t}^{t}\right)+C_{b}^{t}\delta f_{ib}^{b}$$
where $\delta f_{ib}^{b}=\nabla^{b}+w_{a}^{b}$ is the accelerometer assembly error including accelerometer bias $\nabla^{b}$ and accelerometer noise $w_{a}^{b}$.

The gyro and accelerometer biases can be modelled by a first-order Markov process as
(20)$${\dot{\varepsilon }}^{b}=-\frac{1}{\tau_{g}}\varepsilon^{b}+w_{Mg},{\dot{\nabla }}^{b}=-\frac{1}{\tau_{a}}\nabla^{b}+w_{Ma}$$
where $\tau_{g}$ and $\tau_{a}$ are the correlation time of the Markov process and $w_{Mg}$ and $w_{Ma}$ represent the zero-mean Gaussian white noise.

The SINS position error equation projected in the *t* frame is as follows:(21)$$\begin{array}{l} \delta {\dot{\theta }}^{t}=-\omega_{e^{\prime }t}^{t}\times \delta \theta^{t}+\delta \omega_{e^{\prime }t}^{t}\\ \delta {\dot{h}}^{t}=\delta v_{U}^{t} \end{array}$$
where the position error angle and the height error are adopted to describe position error. Moreover, perturbing Equation (3) and rearranging both sides of it yields
(22)$$\delta \theta^{t}={\left[\begin{array}{ccc} -\delta L^{t} & \cos L^{t}\delta \lambda^{t} & \sin L^{t}\delta \lambda^{t} \end{array}\right]}^{T}$$

Equation (22) demonstrates that the elements of the position error angle vector are linearly related:(23)$$\sin L^{t}\delta \theta_{N}^{t}-\cos L^{t}\delta \theta_{U}^{t}=0$$

Therefore, only two error angles are required to describe the error in the position DCM. In this paper, $\delta \theta_{E}^{t}$ and $\delta \theta_{N}^{t}$ are selected, as Equation (18) shows.

For simplicity, the DVL error equation includes the installation error equation and the scale factor error equation. The output of the DVL in the *b* frame can be expressed as follows:(24)$${\tilde{v}}_{DVL}^{d}={\tilde{C}}_{d}^{b}\left(1+\delta k\right)\left(v^{d}+\delta v^{d}\right)$$
with
(25)$${\tilde{C}}_{d}^{b}=\left(I-\left[\varsigma^{b}\times \right]\right)C_{d}^{b}$$
where ${\tilde{C}}_{d}^{b}$ is the estimated installation error matrix, $\varsigma^{b}={\left[\begin{array}{ccc} \delta \alpha & 0 & \delta \gamma \end{array}\right]}^{T}$ is the installation error of the DVL relative to the AUV’s body frame, $C_{d}^{b}$ is the true installation error matrix, δk is the scale factor error of the DVL, $v^{d}$ is the true AUV velocity projected in the *d* frame, and $\delta v^{d}$ is the DVL measurement noise. Since the projection of forward velocity measured by the DVL does not receive the effect of the roll angle, the roll angle error can be ignored. The DVL error parameters can be taken as constants, namely,
(26)$$\delta \dot{k}=0,\delta \dot{\alpha }=0,\delta \dot{\gamma }=0$$

At this point, the system error states are listed as follows:(27)$$x\left(t\right)={\left[\phi_{E}^{t},\phi_{N}^{t},\phi_{U}^{t},\delta v_{E}^{t},\delta v_{N}^{t},\delta v_{U}^{t},\delta \theta_{E}^{t},\delta \theta_{N}^{t},\delta h^{t},\varepsilon_{x}^{b},\varepsilon_{y}^{b},\varepsilon_{z}^{b},\nabla_{x}^{b},\nabla_{y}^{b},\nabla_{z}^{b},\delta k,\delta \alpha ,\delta \gamma \right]}^{T}$$

The system dynamic model is written as
(28)$$\dot{x}\left(t\right)=F\left(t\right)x\left(t\right)+G\left(t\right)w\left(t\right)$$
where F(t) is the system state matrix, G(t) is the system noise matrix, and w(t) is system noise. These matrices can be determined by Equations (15), (19)–(21), and (26).

### 3.2. Measurement Model Design

The difference between the SINS velocity output and the DVL velocity output is used as the Kalman filter measurement vector, which can be expressed as
(29)$${\tilde{v}}_{SINS}^{t}-{\tilde{v}}_{DVL}^{t}=v^{t}+\delta v^{t}-{\tilde{C}}_{b}^{t}{\tilde{C}}_{d}^{b}\left(1+\delta k\right)\left(v^{d}+\delta v^{d}\right)$$
where ${\tilde{v}}_{SINS}^{t}$ is the SINS velocity computed value (which is the sum of true velocity $v^{t}$ and velocity error $\delta v^{t}$) and ${\tilde{C}}_{b}^{t}$ is the computed attitude DCM, which can be written as
(30)$${\tilde{C}}_{b}^{t}=\left(I-\left[\phi^{t}\times \right]\right)C_{b}^{t}$$

Substituting Equations (25) and (30) into Equation (29) yields
(31)$${\tilde{v}}_{INS}^{t}-{\tilde{v}}_{DVL}^{t}=-\left[v^{t}\times \right]\phi^{t}+\delta v^{t}-v^{t}\delta k-\left[v^{t}\times \right]C_{b}^{t}\varsigma^{b}$$

The measurement model can be written as follows:(32)$$z\left(t\right)=\left[\begin{array}{c} {\tilde{v}}_{E,SINS}^{t}-{\tilde{v}}_{E,DVL}^{t}\\ {\tilde{v}}_{N,SINS}^{t}-{\tilde{v}}_{N,DVL}^{t}\\ {\tilde{v}}_{U,SINS}^{t}-{\tilde{v}}_{U,DVL}^{t} \end{array}\right]=H\left(t\right)x\left(t\right)+\upsilon \left(t\right)$$
where z(t) is the measurement vector, H(t) is the measurement matrix whose components can be determined according to Equation (32), and υ(t) is measurement noise.

## 4. Robust Kalman Filter Algorithm

Under the assumptions of Gaussian-distributed system noise and measurement noise, the standard Kalman filter is the optimal choice as its error is the minimal mean squared and it is unbiased and consistent. However, if the system noise or measurement noise is a non-Gaussian distribution, the Kalman filter performance is inevitably degraded, especially regarding the gross error of measurement. In actual application environments, especially in the harsh environment of the polar region, the measurement noise of the DVL will be affected by the environment, and its working state will not stable. There is occasional gross error in the DVL output. These conditions do not meet the assumptions of the standard Kalman filter, thus the normal working state of the integrated navigation filter will be affected. Aiming at solving this problem, this section introduces the Mahalanobis distance of the innovation vector as a criterion to adjust measurement noise variance.

### 4.1. Standard Kalman Filter Algorithm

The Kalman filter is the most commonly used information fusion method in the integrated navigation systems of AUVs. Based on the system dynamic model and measurement model introduced in Section 3.1, the basic assumptions of the Kalman filter regarding system noise and measurement noise are as follows:(33)$$E\left[w_{k}w_{j}^{T}\right]=Q_{k}\delta_{kj}\begin{array}{c} , \end{array}E\left[\upsilon_{k}\upsilon_{j}^{T}\right]=R_{k}\delta_{kj}\begin{array}{c} , \end{array}E\left[w_{k}\upsilon_{j}^{T}\right]=0$$
where $\delta_{kj}$ is the Kronecker delta function (its value is 1 when *k* = *j* and 0 otherwise), $Q_{k}$ is the system noise covariance matrix, and $R_{k}$ is the measurement noise covariance matrix. Both system noise and measurement noise are modeled as zero-mean Gaussian white noise. Generally, the Kalman filter process can be divided into time update and measurement update.

The time update process of the Kalman filter in discrete forms begins with the one-step prediction of system state, which is as follows:(34)$${\hat{X}}_{k/k-1}=\Phi_{k/k-1}{\hat{X}}_{k-1}$$

The covariance matrix of the one-step prediction is as follows:(35)$$P_{k/k-1}=\Phi_{k/k-1}P_{k-1}\Phi_{k/k-1}^{T}+\Gamma_{k-1}Q_{k-1}\Gamma_{k-1}^{T}$$
where ${\hat{X}}_{k-1}$ represents state estimation at the epoch *k*−1, ${\hat{X}}_{k/k-1}$ is the one-step prediction of state estimation, $\Phi_{k/k-1}$ is the state transition matrix, $P_{k/k-1}$ is the one-step prediction of covariance matrix, $P_{k-1}$ is the covariance matrix of state estimation at the epoch *k*−1, and $\Gamma_{k-1}$ is the system noise matrix.

If the DVL output is obtained, its one-step prediction can be used to form an innovation vector, which is as follows:(36)$${\tilde{Z}}_{k}=Z_{k}-H_{k}{\hat{X}}_{k/k-1}$$
where ${\tilde{Z}}_{k}$ is the innovation vector, $Z_{k}$ is the measurement vector at epoch *k*, $H_{k}$ is the measurement matrix, and $Z_{k/k-1}=H_{k}{\hat{X}}_{k/k-1}$ is the measurement one-step prediction. The Kalman filter gain matrix is represented as follows:(37)$$K_{k}=P_{k/k-1}H_{k}^{T}{\left(H_{k}P_{k/k-1}H_{k}^{T}+R_{k}\right)}^{-1}$$
where $P_{{\tilde{Z}}_{k}}=H_{k}P_{k/k-1}H_{k}^{T}+R_{k}$ is the covariance of the innovation vector ${\tilde{Z}}_{k}$.

The state estimation at epoch *k* is as follows:(38)$${\hat{X}}_{k}={\hat{X}}_{k/k-1}+K_{k}{\tilde{Z}}_{k}$$

The covariance matrix at epoch *k* can be represented as follows:(39)$$P_{k}=\left(I-K_{k}H_{k}\right)P_{k-1}{\left(I-K_{k}H_{k}\right)}^{T}+\Gamma_{k-1}Q_{k-1}\Gamma_{k-1}^{T}$$

The standard Kalman filter process is shown in Figure 2.

### 4.2. Robust Kalman Filter Algorithm

The standard Kalman filter is optimal if the DVL measurement noise is a zero-mean Gaussian distribution. However, the harsh polar environment adverse influences DVL performance, which causes gross error and thick-tailed noise to appear [20]. These conditions do not meet the assumptions of the standard Kalman filter and will thus affect the normal working state of the filter. Generally, by adjusting the measurement noise covariance matrix according to some criterion, the performance of the Kalman filter can be improved. The notation of the Mahalanobis distance is introduced here, which denotes the distance of two different vectors.
(40)$$M\left(X,Y\right)=\sqrt{{\left(X-Y\right)}^{T}\Sigma^{T}\left(X-Y\right)}$$
where Σ is covariance matrix. In the standard Kalman filter, the Mahalanobis distance between the measurement vector $Z_{k}$ and its one-step prediction $Z_{k/k-1}$ is defined as Equation (41), which is used as the criterion for judging the abnormal value of the measurement [19]:(41)$$M_{k}^{2}={\left(\sqrt{{\tilde{Z}}_{k}{}^{T}{\left(H_{k}P_{k/k-1}H_{k}^{T}+R_{k}\right)}^{-1}{\tilde{Z}}_{k}}\right)}^{2}$$
where $M_{k}$ denotes Mahalanobis distance at epoch *k*. In fact, if the basic assumptions of the standard Kalman filter are satisfied, $M_{k}^{2}$ should be chi-square distributed with degree of freedom *m*, which is the dimensional measurement vector. Therefore, the chi-square test can be used to determine if $Z_{k}$ is an outlier. According to the given statistical threshold $\chi_{\alpha }$ and significance level α, if $M_{k}^{2}\leq \chi_{\alpha }^{2}$, $Z_{k}$ is considered as a normal measurement; however, if $M_{k}^{2}>\chi_{\alpha }^{2}$, $Z_{k}$ is considered as an outlier, and the measurement noise is inflated as follows to achieve robust estimation:(42)$${\tilde{R}}_{k}=\lambda_{k}R_{k}$$
where ${\tilde{R}}_{k}$ denotes the inflated measurement noise covariance and $\lambda_{k}$ the inflated factor of $R_{k}$. Substituting Equation (42) into Equation (41) yields
(43)$${\tilde{M}}_{k}^{2}={\left(\sqrt{{\tilde{Z}}_{k}{}^{T}{\left(H_{k}P_{k/k-1}H_{k}^{T}+\lambda_{k}R_{k}\right)}^{-1}{\tilde{Z}}_{k}}\right)}^{2}$$

At this point, ${\tilde{M}}_{k}^{}$ can be viewed as a function of $\lambda_{k}$. By the choice of $\lambda_{k}$, ${\tilde{M}}_{k}^{}$ should meet the normal value judgment
(44)$$f\left(\lambda_{k}\right)={\left(\sqrt{{\tilde{Z}}_{k}{}^{T}{\left({\hat{P}}_{k/k-1}+\lambda_{k}R_{k}\right)}^{-1}{\tilde{Z}}_{k}}\right)}^{2}-\chi_{\alpha }^{2}\leq 0$$
where ${\hat{P}}_{k/k-1}=H_{k}P_{k/k-1}H_{k}^{T}$. $\lambda_{k}$ can be solved by Newton’s iterative method, which is given by
(45)$$\lambda_{k}\left(i+1\right)=\lambda_{k}\left(i\right)+\frac{f\left(\lambda_{k}\left(i\right)\right)}{f^{\prime }\left(\lambda_{k}\left(i\right)\right)}$$
where *i* is the number of iterations. The initial value of $\lambda_{k}$ is selected as 1.

According to Equation (45), the iterative calculation process of $\lambda_{k}$ is expressed as follows:(46)$$\lambda_{k}\left(i+1\right)=\lambda_{k}\left(i\right)+\frac{{\tilde{M}}_{k}^{2}\left(i\right)-\chi_{\alpha }^{2}}{{\tilde{Z}}_{k}^{T}{\left({\hat{P}}_{k/k-1}\right)}^{-1}R_{k}{\left({\hat{P}}_{k/k-1}\right)}^{-1}{\tilde{Z}}_{k}}$$
and the iterative process terminates when ${\tilde{M}}_{k}^{2}\left(i\right)\leq \chi_{\alpha }^{2}$.

After solving for $\lambda_{k}$, the adjust filter gain matrix can be expressed as follows:(47)$${\tilde{K}}_{k}=P_{k/k-1}H_{k}^{T}{\left(H_{k}P_{k/k-1}H_{k}^{T}+\lambda_{k}R_{k}\right)}^{-1}$$

The robust state estimation can be updated based on the above mentioned ${\tilde{K}}_{k}$. To sum up, the algorithm of the robust Kalman filter is shown in Figure 3.

## 5. Experimental Result and Discussion

To verify the effectiveness of the proposed algorithm in this paper, a ship experiment and semi-physical simulation experiment were carried out. Since the transverse frame is also applicable at mid-latitudes, the effectiveness of the proposed algorithm is firstly verified through a ship experiment. The collected raw data of the SINS and DVL is then converted to the polar regions to further verify the effectiveness of the algorithm using the semi-physical simulation method proposed in [8].

### 5.1. Ship Experiment

The trial ship was equipped with a DVL-aided SINS and a GNSS, which was used as the reference equipment. The corresponding equipment parameters are listed in Table 1. The sailing area was a section of the Yangtze River. The velocity measured by the DVL was relative to the ground. The ship trajectory is shown in Figure 4a,b, and the coordinates of the starting position were [110.98° E, 30.96° N]. The ship experiment lasted about 6 h. The first 900 s of the experiment was for initial SINS alignment, and the ship returned after about 3.3 h. The velocity measured by the DVL is projected in the *d*-frame, and it can be converted to the *b*-frame with the DVL installation error parameters. As shown in Figure 5, there are some outliers in the DVL velocity output because of its unstable working condition. In addition, the unknown environment also affects the working conditions of the DVL, and its measurement accuracy decreases as a result.

However, the outliers in the DVL outputs are difficult identify with the standard Kalman filter. If they are used as a measurement vector by the standard Kalman filter, the performance will inevitably be degraded. To verify the effectiveness of the proposed algorithm, three groups of experiments were carried out for comparison, and the navigation frame of the three groups of experiments was the transverse frame in all cases. Moreover, for the sake of convenience, all the experimental results were converted to the traditional local-level geographic frame for presentation. The three groups of comparison experiments are listed as follows:Group 1 directly used the DVL velocity output as the measurement vector, and the standard Kalman filter (KF) was performed for data fusion.Group 2 also used the standard Kalman filter for data fusion, but only a time update was performed when the DVL worked abnormally.Group 3 used a robust Kalman filter (RKF) for data fusion and did not remove the outliers of DVL outputs.

The measurement noise variance used by the standard Kalman filter and robust Kalman filter is shown in Figure 6. It can be seen that the robust Kalman filter can adjust the measurement noise covariance value adaptively according to the DVL measurement value. However, the standard Kalman filter only uses a constant measurement noise covariance value. Taking GNSS-aided SINS velocity as the benchmark, comparison of the velocity errors of the three groups is shown in Figure 7, which can demonstrate the influence of the outliers on the integrated navigation accuracy. Besides the east velocity error and the north velocity error, the radial velocity error is also calculated, which is the vector sum of the east velocity error and north velocity error. Since the upward velocity measured by the DVL is not accurate, the trial ship’s upward velocity is considered to be 0 m/s in the experiments. As can be seen from Figure 7, the velocity errors of Group 1 are drastically affected by the outliers and, in the case of abnormal DVL output appearing, the wrong velocity values affect the normal operation of the integrated navigation system. The maximum radial velocity error of the three groups was 7.09 m/s, 0.91 m/s, and 0.87 m/s, respectively. The root mean square error (RSME) of the radial velocity is shown in Figure 8, which was calculated every 10 s. The time-averaged RMSE (TARMSE) of the three groups was 0.24 m/s, 0.13 m/s, and 0.073 m/s, respectively. Group 2 and Group 3 are almost unaffected by the DVL outliers. Note that Group 2 can only be used as a post-processing result due to the fact that outliers are artificially removed.

The comparison of positioning errors from the three groups is shown in Figure 9. It can be seen that the effect of velocity outliers on positioning accuracy is devastating. The positioning error of Group 1 oscillates dramatically due to the presence of DVL outliers, and the maximum radial positioning error is about 457 m. In addition, the outliers cause the estimated trajectory of Group 1 to gradually deviate from the true trajectory. Although the positioning accuracy of Group 2 is improved by removing the outliers and its maximum radial positioning error is about 117.7 m, it is obvious in Figure 9 that the positioning error oscillates significantly at the turn around. This is due to the rapid change in the ship’s attitude that leads to the inaccurate DVL velocity output. During this period, only the pure inertial navigation is performed, leading to the accumulation of navigation errors. This also illustrates the limitation of the standard Kalman filter, which cannot adaptively modify the measurement noise covariance value to eliminate the adverse impact of inaccurate DVL velocity. As expected, the maximum radial positioning error of Group 3 is about 81.5 m, which is the minimum value among the three groups in the experiments. The RMSE of the position is shown in Figure 10, which was calculated every 10 s. The TARMSE of the three Groups was 326.89 m, 53.48 m, and 40.22 m, respectively. Group 3 improved by 87.7% compared to Group 1 and improved by 24.8% compared to Group 2. Since the measurement noise covariance value is adaptively modified according to the DVL measurement value, the robust Kalman filter significantly improves positioning accuracy.

Although outliers in the DVL output can be artificially removed through data post-processing, the identification of outliers using a standard Kalman filter is difficult in the actual navigation system. However, the robust Kalman filter can effectively solve this problem. This experiment verifies the effectiveness of the robust Kalman filter in DVL-aided SINSs in the transverse frame.

### 5.2. Semi-Physical Simulation Experiment

Due to the difficulty of conducting experiments in the polar regions, the semi-physical simulation method proposed in [8] is adopted to generate the raw data of SINSs and DVLs at high latitudes, which can then be used to verify the algorithm proposed in this paper. The experimental data collected at mid-latitudes were converted to simulate the trial ship condition in the polar region, and the converted trajectory is shown in Figure 11. The simulated trial ship trajectory is near the 80 °N. The semi-physical simulation method used in this paper keeps the attitude and velocity of the trial ship relative to the ground unchanged and only changes the position relative to the earth. Thus, the DVL measurement is still valid in the experiment. Moreover, the maneuvering situation of the trial ship can be maintained to the greatest extent possible. However, compared to the mid-latitudes, the increased latitude makes the longitude variation region bigger in the case of identical trajectories. Figure 12 shows the simulated DVL velocity outputs. As mentioned above, the semi-physical simulation method adopted in this paper can maintain an unchanged DVL velocity measurement. As a result, the outliers still exist in the DVL outputs, and the adverse effect of them should be removed by the filter algorithm.

Three groups of experiments were conducted for comparison that were the same as the experiments at mid-latitudes. The navigation frame used was the *t*-frame. Comparison of the velocity errors for the three groups of experiments is shown in Figure 13. As shown, the velocity errors at high latitudes are almost the same as those at mid-latitudes due to the fact that the DVL velocity measurement remains unchanged. The maximum radial velocity errors of the three groups were 7.11 m/s, 0.98 m/s, and 0.84 m/s, respectively. The RMSE of radial velocity is shown in Figure 14, which was calculated every 10 s. The TARMSE of the three groups was 0.24 m/s, 0.095 m/s, and 0.078 m/s, respectively. The results are consistent with the mid-latitude regions. Importantly, the standard Kalman filter cannot resist the adverse effect of DVL outliers. The robust Kalman filter can solve this problem well. The comparison of positioning errors is shown in Figure 15. It can be seen that the positioning errors of the three groups of experiments all increased as the latitudes rose. The reason for this is that initial alignment accuracy decreases at high latitudes, and the performance of the DVL-aided SINS is therefore inevitably influenced. The maximum radial positioning error of the three groups of experiments was 1289.0 m, 328.7 m, and 250.3 m, respectively. The RMSE of positions was calculated every ten seconds and is shown in Figure 16. The TARMSE of the three groups was 880.13 m, 170.46 m, and 141.69 m, respectively. The positioning accuracy of Group 3 was improved by 83.9% and 16.88% compared to Group 1 and Group 2. Moreover, at high latitudes, the adverse effect of the DVL outliers is even greater than it was for mid-latitudes. The reason for this is that observability of the system error states is decreased, which makes these estimations susceptibly influenced by the filter measurement. Through semi-physical simulation experiments, the effectiveness of the algorithm in the polar region is proven. The robust Kalman filter algorithm can resist the adverse effect of the DVL outliers and effectively improve the positioning accuracy of DVL-aided SINSs.

## 6. Conclusions

In order to address the AUV polar navigation issue, a polar robust Kalman filter algorithm for DVL-aided SINSs based on the earth ellipsoidal model is proposed in this paper. The transverse local-level geographic frame is defined as the navigation frame in the polar region, and the related navigation parameters are redefined. Based on the spherical earth assumption, the system dynamic model and measurement model of the DVL-aided SINS are respectively designed, and the error related to the earth parameter can be avoided. Further, a robust Kalman filter with Mahalanobis distance as a criterion is proposed to resist the influence of DVL outliers on the integrated navigation results and adaptively estimate the measurement noise covariance value, whereby the Kalman filter gain can be modified to weight the measurement. A trial ship experiment and semi-physical simulation experiment were carried out to verify the effectiveness of the proposed algorithm. The results demonstrate that the proposed algorithm can effectively resist the influence of DVL outliers and improve positioning accuracy.

## Figures and Tables

**Figure 1 sensors-22-07879-f001:**
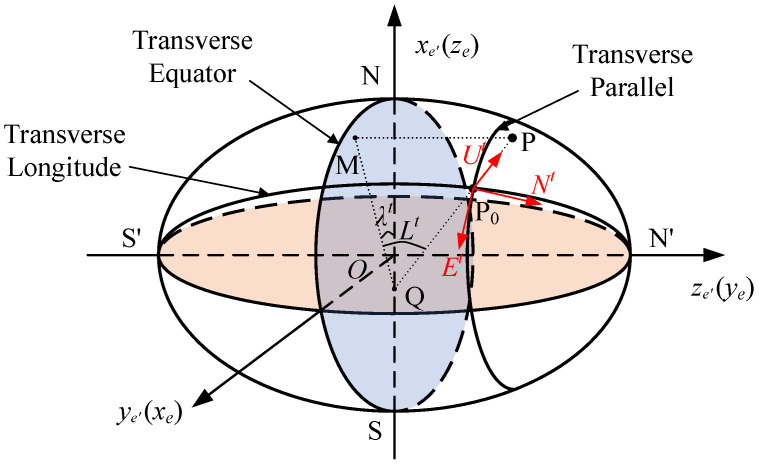
Definition of the transverse frame.

**Figure 2 sensors-22-07879-f002:**
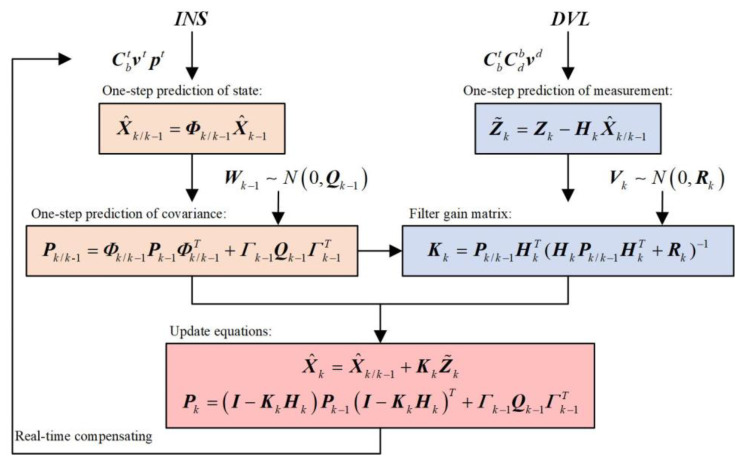
The standard Kalman filter flow diagram.

**Figure 3 sensors-22-07879-f003:**
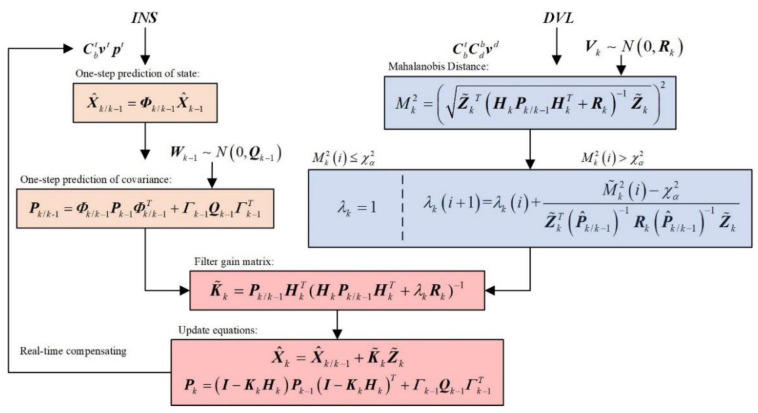
The robust Kalman filter flow diagram.

**Figure 4 sensors-22-07879-f004:**
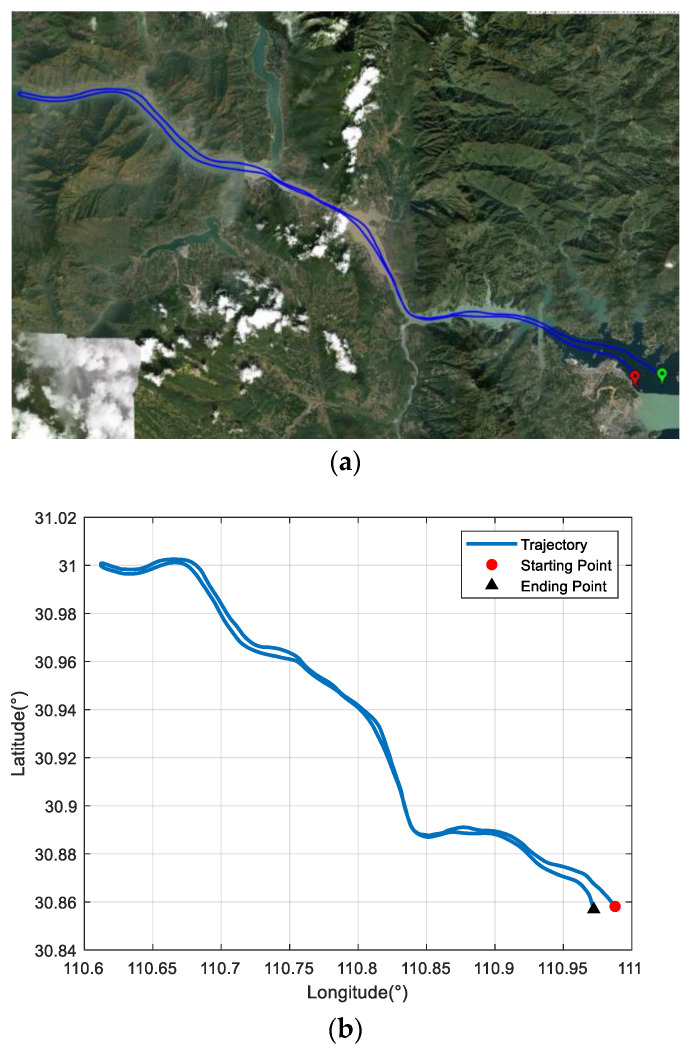
Ship trajectory at mid-latitudes: (**a**) planned route in Google Maps; (**b**) actual navigation trajectory.

**Figure 5 sensors-22-07879-f005:**
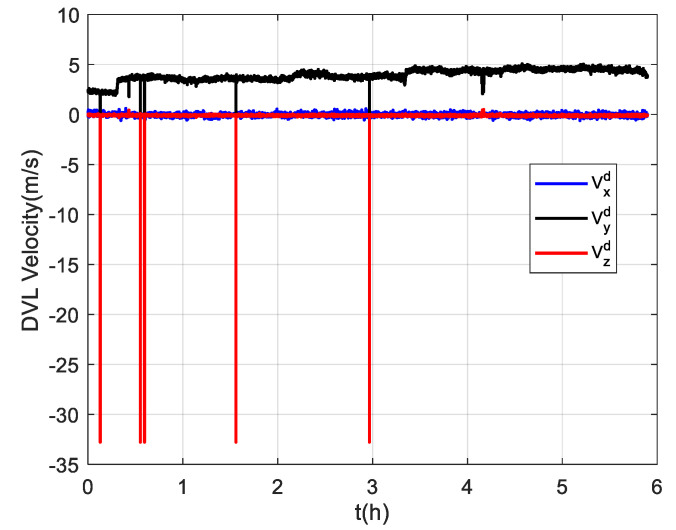
The velocity outputs of the DVL.

**Figure 6 sensors-22-07879-f006:**
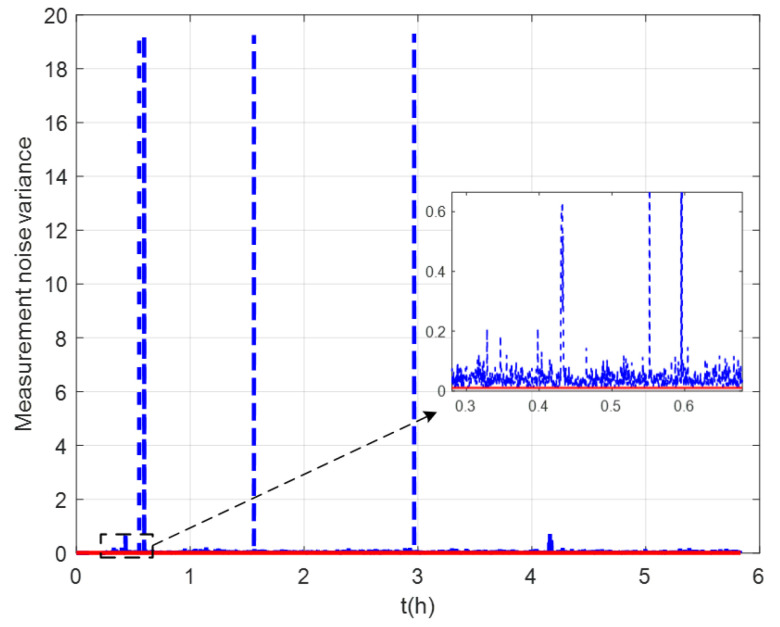
Values of measurement noise covariance.

**Figure 7 sensors-22-07879-f007:**
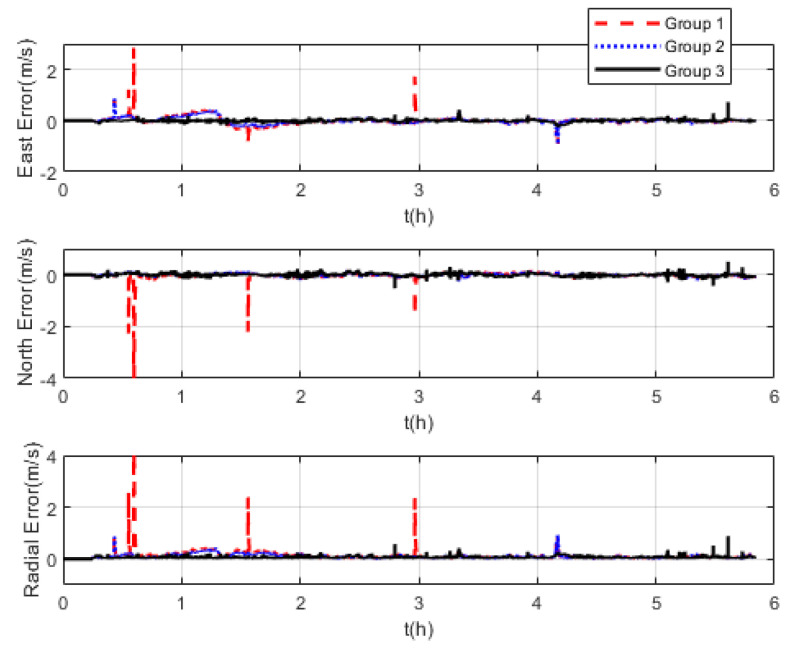
Comparison of velocity errors at mid−latitudes.

**Figure 8 sensors-22-07879-f008:**
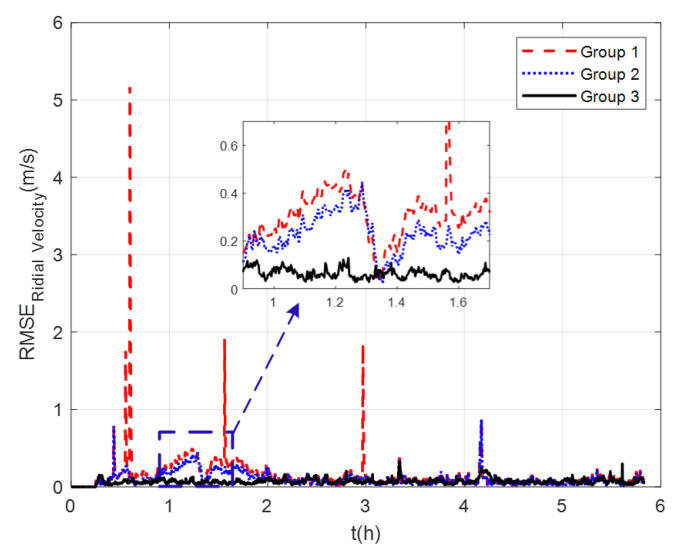
RSME of radial velocity at mid−latitudes.

**Figure 9 sensors-22-07879-f009:**
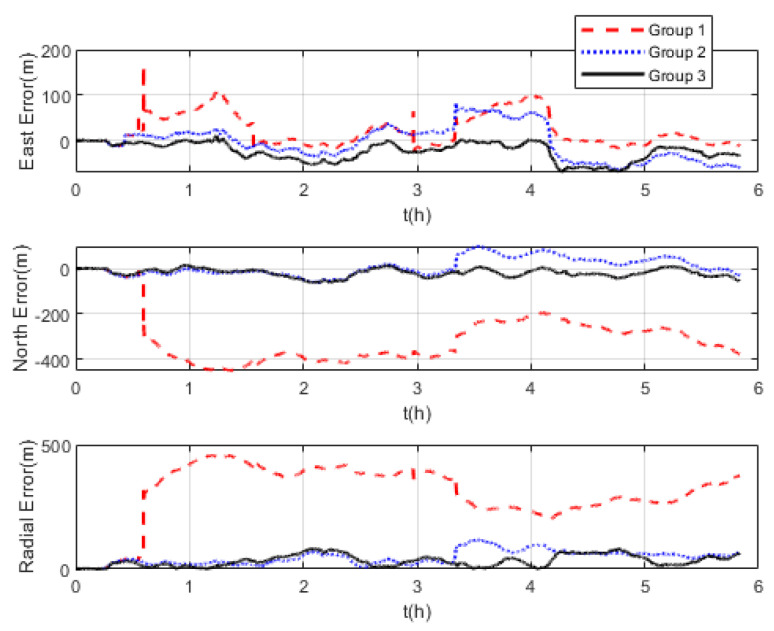
Comparison of positioning errors at mid−latitudes.

**Figure 10 sensors-22-07879-f010:**
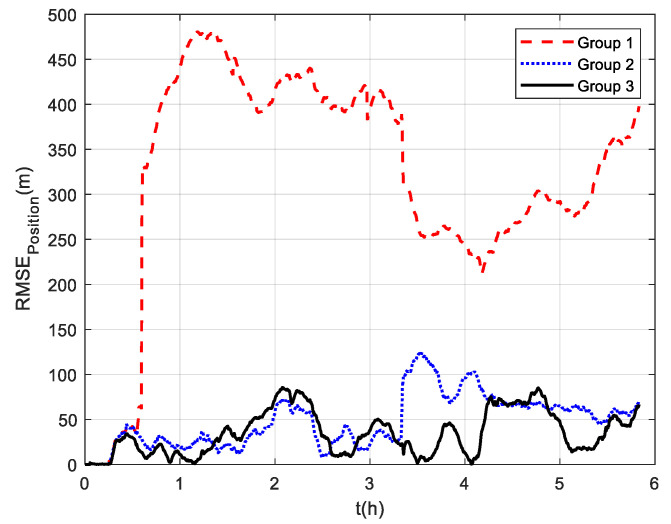
RSME of positions at mid−latitudes.

**Figure 11 sensors-22-07879-f011:**
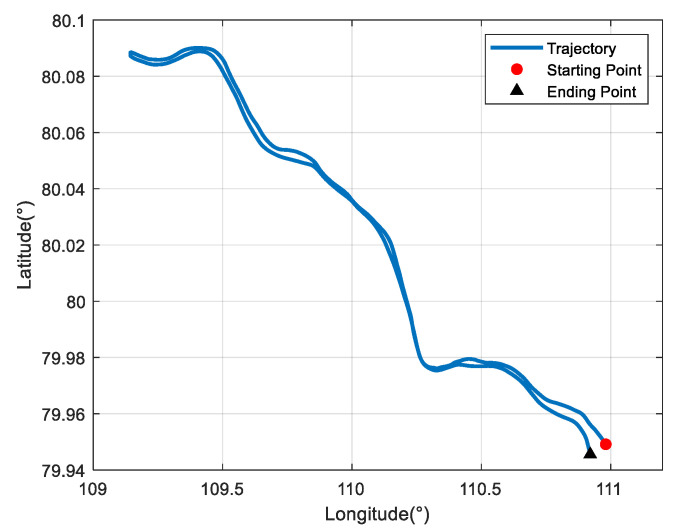
Ship trajectory at high latitudes.

**Figure 12 sensors-22-07879-f012:**
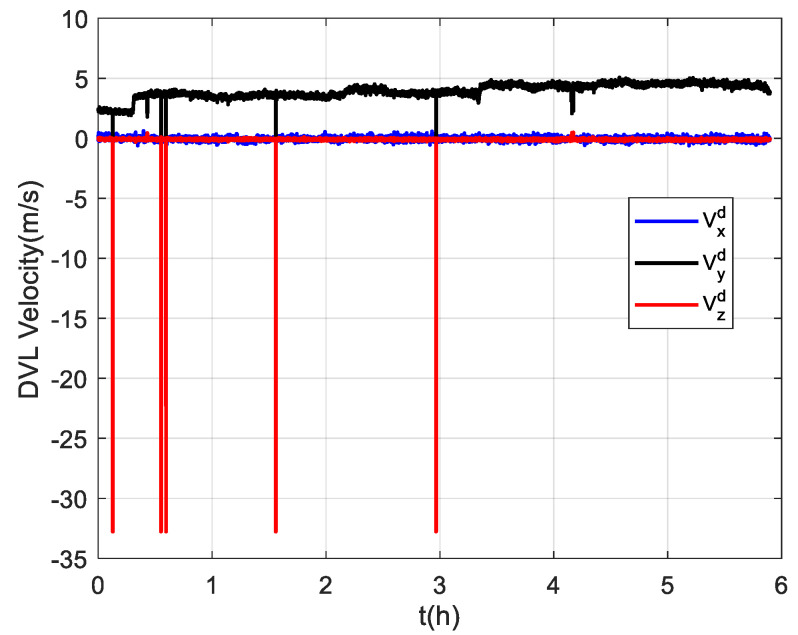
The simulated velocity outputs of the DVL at high latitudes.

**Figure 13 sensors-22-07879-f013:**
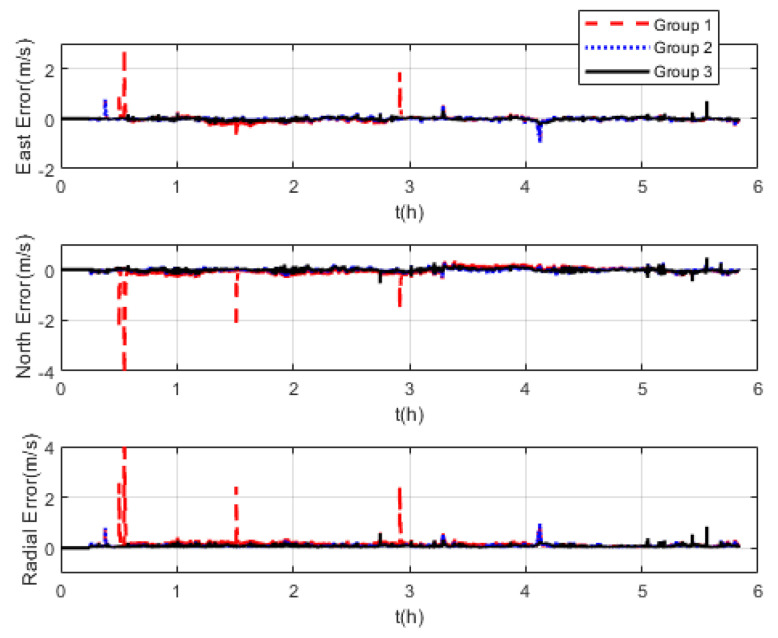
Velocity errors at high latitudes.

**Figure 14 sensors-22-07879-f014:**
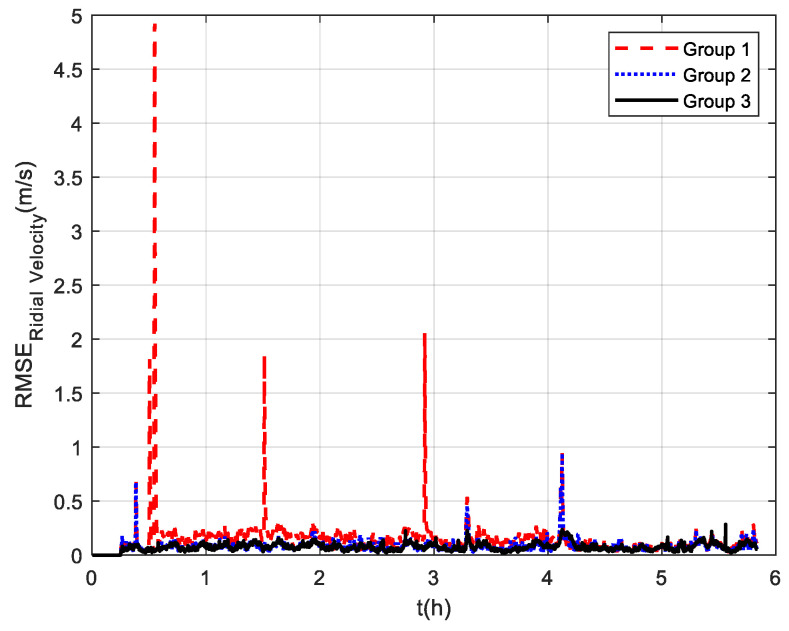
RMSE of radial velocity at high latitudes.

**Figure 15 sensors-22-07879-f015:**
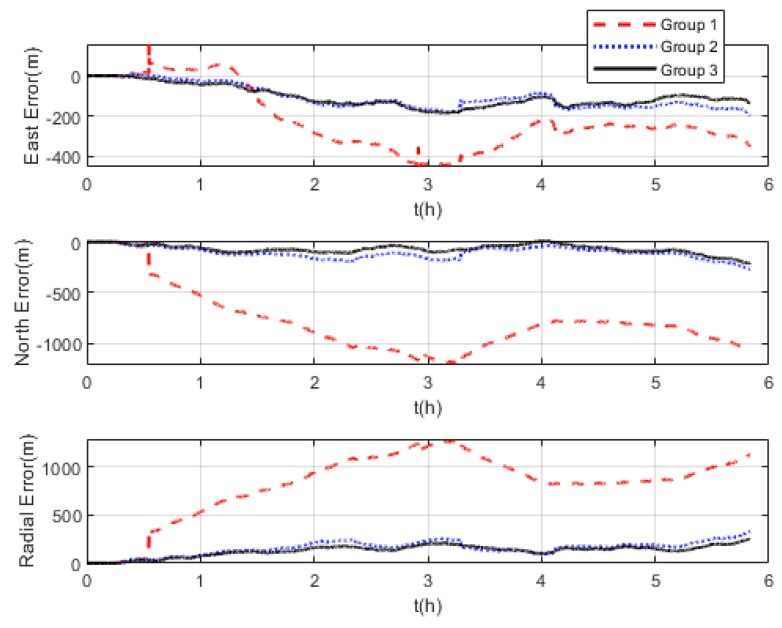
Positioning errors at high latitudes.

**Figure 16 sensors-22-07879-f016:**
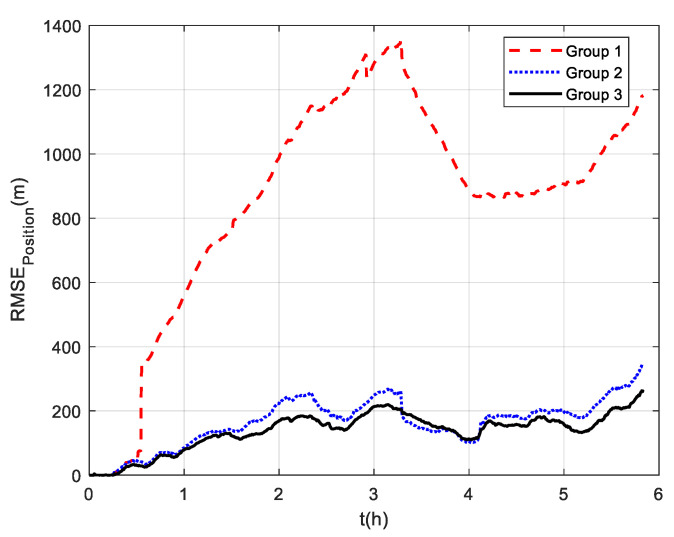
RSME of position at high latitudes.

**Table 1 sensors-22-07879-t001:** The parameters of the measuring instrument.

Measuring Instrument	Parameters	
SINS	Gyro drifts stability Accelerometer bias stability	<0.02 °/h <50 μg
DVL	Accuracy	0.5% V ± 5 mm/s
GNSS	Positioning accuracy	<5 m

## Data Availability

Not applicable.

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
