# Peer review of "A Polar Robust Kalman Filter Algorithm for DVL-Aided SINSs Based on the Ellipsoidal Earth Model"

_sensors, 2022, doi:10.3390/s22207879_

Round 1

Reviewer 1 Report

In this manuscript, the authors claim “Polar Robust Kalman Filter Algorithm for DVL-aided SINS Based on Earth Ellipsoidal Model.” A transverse inertial navigation mechanism based on earth ellipsoidal model is designed in this manuscript. Then, a robust Kalman filter algorithm based on Mahalanobis distance is used to adaptively estimate the measurement noise covariance, and hence the Kalman filter gain can be modified to weight the measurement. However, there are several points the authors need to address:

1)    From the manuscript, the authors give the detailed analysis of the transverse model for the SINS/DVL integration. However, this is the basic principle, it is need to simplified. 

2)    Why the Mahalanobis distance is used to design the robust Kalman filter? There are many other methods, such as Huber’s method and Student-t method.

3)    The ship tests are not sufficient; I think the other robust Kalman filter is needed to add for comparison.

4)    The figures are not clearly, such as figure 9 and 13.

5)    Why need the semi-physical test since the ship test is designed.

Reviewer 2 Report

Overall, this is an interesting paper. I recommend following changes.

1. Enrich the literature review in your work.

2. Highlight the innovative contributions of the proposed methodology in the Introduction Section.

3. Improve the comprehension of the discussion related to the proposed scheme.

4. Comment on the originality / novelty of the said technique.

5. Include a concise quantitative and qualitative comparative analysis of the experimental results.

6. Check for grammatical and spelling mistakes.

Reviewer 4 Report

A DVL-aided SINS navigation algorithm to address the difficulties of navigation in polar regions is widely studied in the manuscript. The effectiveness of the algorithm is verified using ship experiments and semi-physical simulation experiments. The method has practical engineering significance and innovation. However, there are also some minor comments.

1. In equation (5), it is proposed to supplement the conversion relationship between transverse latitude, transverse longitude, and the elements in equation (3), considering that the position error representation later in this paper is based on the position error angle.

2. Why the roll angle error is not included in the installation error angles? It is suggested to supplement in the paper.

3. Equation (30) should have the same form as equation (25).

4. The meaning of k for in equation (41) should be explained clearly.

5. In the experiment, the velocity measured by DVL is relative to the water or to the ground? It should be stated in the paper.

6. The figure (7), figure (9), figure (13), and figure (15) would be more readable by adding grid lines.

7. Can the real-time performance of the Robust Kalman filter algorithm proposed in the paper be guaranteed?

Round 2

Reviewer 1 Report

Thank you very much for your revision, I think this version can be accepted.